# Molecular Epidemiology of Methicillin-Resistant *Staphylococcus aureus* among Patients Diagnosed with Surgical Site Infection at Four Hospitals in Ethiopia

**DOI:** 10.3390/antibiotics12121681

**Published:** 2023-11-29

**Authors:** Seble Worku, Tamrat Abebe, Berhanu Seyoum, Ashenafi Alemu, Yidenek Shimelash, Marechign Yimer, Alemseged Abdissa, Getachew Tesfaye Beyene, Göte Swedberg, Adane Mihret

**Affiliations:** 1Department of Microbiology, Immunology and Parasitology, School of Medicine, College of Health Sciences, Addis Ababa University, Addis Ababa 1165, Ethiopia; tamrat.abebe@aau.edu.et (T.A.); adane.mihret@ahri.gov.et (A.M.); 2Department of Medical Laboratory Science, College of Health Sciences, Debre Tabor University, Debre Tabor P.O. Box 272, Ethiopia; 3Bacterial and Viral Diseases Research Directorate, Armauer Hansen Research Institute, Addis Ababa 1165, Ethiopia; berhanu.seyoum@ahri.gov.et (B.S.); ashenafialemu07@gmail.com (A.A.); marechign01@yahoo.com (M.Y.); alemseged.abdissa@ahri.gov.et (A.A.); getachew.tesfaye@ahri.gov.et (G.T.B.); 4Debre Tabor Comprehensive Specialized Hospital, Debre Tabor P.O. Box 272, Ethiopia; yidenek.shim2008@gmail.com; 5Department of Medical Biochemistry and Microbiology, Uppsala University, 750 08 Uppsala, Sweden; gote.swedberg@imbim.uu.se

**Keywords:** surgical site infection, methicillin-resistant *Staphylococci*, molecular epidemiology, antimicrobial resistance, Ethiopia

## Abstract

Methicillin-resistant *Staphylococcus aureus* (MRSA) is a common cause of severe surgical site infections (SSI). The molecular epidemiology of MRSA is poorly documented in Ethiopia. This study is designed to determine the prevalence of MRSA and associated factors among patients diagnosed with SSI. A multicenter study was conducted at four hospitals in Ethiopia. A wound culture was performed among 752 SSI patients. This study isolated *S. aureus* and identified MRSA using standard bacteriology, Matrix-Assisted Laser Desorption/Ionization Time-of-Flight Mass Spectrometry (MALDI-TOF MS), and cefoxitin disk diffusion test. The genes *mecA*, *femA*, *vanA*, and *vanB* were detected through PCR tests. *S. aureus* was identified in 21.6% of participants, with 24.5% of these being methicillin-resistant Staphylococci and 0.6% showing vancomycin resistance. Using MALDI-TOF MS for the 40 methicillin-resistant Staphylococci, we confirmed that 31 (77.5%) were *S. aureus*, 6 (15%) were *Mammaliicoccus sciuri*, and the other 3 (2.5%) were *Staphylococcus warneri*, *Staphylococcus epidermidis*, and *Staphylococcus haemolyticus*. The gene *mecA* was detected from 27.5% (11/40) of Staphylococci through PCR. Only 36.4% (4/11) were detected in *S. aureus*, and no *vanA* or *vanB* genes were identified. Out of 11 *mecA*-gene-positive Staphylococci, 8 (72.7%) were detected in Debre Tabor Comprehensive Specialized Hospital. Methicillin-resistant staphylococcal infections were associated with the following risk factors: age ≥ 61 years, prolonged duration of hospital stay, and history of previous antibiotic use, *p*-values < 0.05. Hospitals should strengthen infection prevention and control strategies and start antimicrobial stewardship programs.

## 1. Introduction

*Staphylococcus aureus* (*S. aureus*) is a Gram-positive coccus that causes significant infections worldwide, including bacteremia, endocarditis, osteomyelitis, and skin and soft tissue infections, due to its easy transmission and commensal nature [1]. Not only *S. aureus* but also coagulase-negative Staphylococci (CoNS), which currently are defined as more than 40 species, are frequently associated with opportunistic human infections. *S. epidermidis* and *S. haemolyticus* are the major species of CoNS frequently isolated from clinical specimens [2]. Furthermore, *Mammaliicoccus sciuri* (previously called *S. sciuri*) is part of the normal flora of goats and camels, and it is a rare opportunistic pathogen in humans [3]. *S. aureus* possesses a unique set of virulence factors, including toxins, enzymes, and metallophores, which enable it to survive extreme conditions, promote tissue colonization, cause systemic infection, *and* evade the host’s immunity [4]. By utilizing metallophores, this bacterium can sequester metal ions from its environment [5]. *S. aureus* infections have previously been treated with beta-lactams, including penicillin and, later, methicillin, as well as sulfonamides, tetracyclines, and macrolides [6]. However, antibiotic-resistant strains of *S. aureus* have developed due to repeated exposure to antibiotics, leading to an increase in methicillin-resistant *S. aureus* (MRSA) infections globally. MRSA is one of the most causative pathogens of surgical site infections (SSIs), and it is a prevalent bacterium that frequently colonizes hospital environments and causes hospital-acquired infections [6] and community-acquired infections [7].

MRSA is characterized by resistance to penicillins, cephalosporins, and carbapenems, with the exception of the new anti-MRSA cephalosporins ceftaroline and ceftobiprole antimicrobial agents [6,8]. The main mechanism of resistance is an altered penicillin-binding protein (PBP2a/c) encoded by the *mecA* gene [1]. The *mecA* gene is regarded as the gold standard for identifying isolates of MRSA, and it is a helpful marker. It is highly conserved in staphylococcal strains and acquired through horizontal gene transfer. *mecA* is carried/located on the mobile genetic element staphylococcal cassette chromosome (SCC) mec, and it codes for the low-affinity PBP2a [9]. Other chromosomal factors, such as the high-level expression of *femA* and *femB*, also seem to be essential for high-level methicillin resistance [10]. The current treatment options for more serious MRSA infections requiring hospitalization include parenteral antimicrobials, such as teicoplanin, tigecycline, linezolid [8], trimethoprim–sulfamethoxazole, doxycycline, daptomycin [6], and vancomycin [8]. However, the majority of MRSA strains are capable of evolution and have acquired resistance to a variety of antibiotics, including those listed above [9,11]. 

Vancomycin resistance in MRSA was first discovered in 1996 in Japan following a few years of commercializing the antibiotic [12]. Vancomycin resistance is acquired through mutations and cell wall modification [12,13] mediated by a *vanA* gene cluster that can be acquired from vancomycin-resistant enterococcus (VRE) [11] through mobile genetic elements like transposonTn1546 [14]. Vancomycin-resistant *S. aureus* (VRSA) infections are treated with antibiotics like tigecycline, quinupristin, daptomycin, ceftobiprole, iclaprim, linezolid, and new glycopeptides (telavancin, oritavancin, and dabavancin) [15].

Globally, the prevalence of VRSA was 16% in Africa, 1% in Europe, 3% in South America, 4% in North America, and 5% in Asia [16]. A systematic review and meta-analysis revealed a highly variable prevalence of VRSA and MRSA in Ethiopian *S. aureus* isolates. The MRSA prevalence ranged from 8.3% to 77.3% (with a pooled prevalence of 32.5%) [17]. In the same way, there was a 5.1% to 44.3% variation in VRSA prevalence. [18]. These days, MRSA is considered a serious threat to public health, and it is one of the pathogens that needs to be treated with high priority. However, the molecular epidemiology of MRSA and VRSA is less well documented in Ethiopia, and published reports on MRSA- and VRSA-causing SSIs are scarce. Furthermore, almost all earlier reports depend on phenotypic laboratory methods. Therefore, in this study, we used the Matrix-Assisted Laser Desorption/Ionization Time-of-Flight Mass Spectrometry (MALDI-TOF MS) technique for the confirmation of bacterial isolates and multiplex polymerase chain reaction (PCR) for the detection of *mecA*, *femA*, *vanA*, and *vanB*. 

## 2. Results

In this study, a total of 752 participants were included. *S. aureus* was isolated from 21.7% (163) of these participants. Following that, a cefoxitin disc diffusion test was used as a substitute marker for oxacillin and other penicillinase-resistant penicillins to ascertain the percentage of MRSA. 

Of all participants, 5.3% (40/752) carried bacteria characterized as MRSA, while among isolates of *S. aureus*, the frequency of MRSA was 24.5% (Table 1). All methicillin-resistant isolates were also tested for vancomycin susceptibility. Except for one isolate (2.5%), all tested isolates for vancomycin were sensitive. 

The age of study participants with MRSA ranged from 8 days to 85 years, with a mean age (±standard deviation) of 35 ± 28.3 years and a median of 30 years, and 58.3% (95) were males. Fifty-nine (36.2%) of the participants received antimicrobial prophylaxis before the procedure, and 47.2% (63) underwent surgeries lasting longer than an hour (Table 2). 

The likelihood of MRSA SSI occurrence was about 3.7 times higher among patients aged ≥ 61 (AOR = 3.729 (1.179–11.791)) compared to those aged ≤ 60. Similarly, the relative risk of MRSA SSI occurrence was about 1.9 times more likely among patients who had a hospital stay ≥ 7 days (AOR = 1.856 (0.688–5.311)). Also, those who had a history of antibiotic use had a 3.7 times higher risk of developing methicillin-resistant Staphylococci infections (AOR = 3.692 (1.059–2.800)) than methicillin-sensitive *S. aureus* (MSSA) SSI. The likelihood of SSI occurrence was about 3.16 times more likely among patients who had antimicrobial prophylaxis during the operation (AOR = 3.066 (1.101–9.392)) than those who had antimicrobial prophylaxis before the operation. All *p*-value < 0.05 (Table 3). 

### 2.1. MALDI-TOF MS Identification of Methicillin-Resistant Staphylococcus Isolates

Of the 40 phenotypic MRSA bacterial isolates, MALDI-TOF MS only identified 77.5% (31/40) as *S. aureus*, while 6 were identified as *M. sciuri*, and the other three as *S. warneri*, *S. epidermidis*, and *S. haemolyticus* (Figure 1). The majority (70%) of methicillin-resistant Staphylococcus isolates were identified from Debre Tabor Comprehensive Specialized Hospital (Figure 2) with 47.5% as *S. aureus*, 15% as *M. sciuri*, 2.5% as *S. warneri*, 2.5% as *S. epidermidis* and 2.5% as *S. haemolyticus.*

### 2.2. PCR amplification of mecA, femA, van A, and vanB 

Detection of *mecA*, *femA*, *van A*, and *vanB* was performed for all MRSA and methicillin-resistant Staphylococci other than *S. aureus* (MRSOSA). The PCR tests revealed that 27.5% (11/40) contained the *mecA* gene, 25% (10/40) were both *mecA*- and *femA*- positive, and 92.5% (37/40) showed the *femA* gene (Figure 3). On the other hand, from all eleven isolates that contained the *mecA* gene only, four were *S. aureus*, whereas five were *M. sciuri*, one was *S. warneri*, and one was *S. haemolyticus*, respectively (Figure 4 and Table 4). Among *S. aureus* isolates, only 12.9% (4/31) carried the *mecA* gene (MRSA), whereas 83.3% (5/6) of *M. sciuri* and both *S. warneri* and *S. haemolyticus* isolates carried *mecA* (Figure 3).

The 11 isolates that contained the *mecA* gene, as shown in Figure 4A,B, were *S. aureus* (lanes 1, 5, 18, and 19), *M. sciuri* (lanes 7, 23, 24, 29, and 31), *S. hemolyticus* (lane 11), and *S. warneri* (lane 12). These were analyzed for *vanA* and *vanB*, and none of these isolates showed *vanA* or *vanB* in the gel electrophoresis (Table 5).

Most of the isolates carrying both the *mecA* and *femA* gene were reported from Debre Tabor (Figure 5). At Debre Tabor Comprehensive Specialized Hospital, 72.7% of *mecA*-positive, 70% of cefoxitin-resistant, and 67.7% of *femA*-positive Staphylococci were discovered (Figure 5).

## 3. Discussion

MRSA is one of the primary bacteria responsible for surgical site infections [19]. The bacteria are human commensals [20], and they can cause a variety of infections, including simple skin and wound infections; they can also infect visceral organs. If not diagnosed and treated properly, many of these illnesses can quickly become life-threatening diseases [1]. 

In our study, among the 752 wound swab samples processed, we detected 21.7% (163/752) of *S. aureus* phenotypically. The present finding is similar to previous studies reported from Jimma Ethiopia (23.6%) [21] and Brazil (20%) [22]. On the other hand, this finding is lower than those of studies conducted in other parts of Ethiopia, such as Dessie (34.5%) [23] and Debre Markos (39.7%) [24]. The variation in prevalence between studies might be due to variations in the study subjects, the conducted time, and the method employed for the detection of *S. aureus* [25]. 

The proportion of MRSA among the isolates based on disc diffusions was 24.5% (40/163). This study’s findings were similar to a previous study conducted in an Indian Hospital (21.7%) [26]. On the other hand, the finding showed higher frequency than earlier studies in Ethiopia from Dessie (9.8%) [23] and Debre Markos (13.2%) [24], but it was below the national pooled prevalence estimate of Ethiopia (32.5%) [17], Addis Ababa (68.4%) [27], Arba Minch (82.3%) [28], and Nigeria (44%) [29]. Variations in MRSA prevalence across countries are influenced by demographics, antibiotic prescription policies, infection prevention and control programs, staff and elderly hygiene education, healthcare system structure, and MRSA diagnostic facilities [30,31].

From those tested for vancomycin resistance, one isolate had a minimum inhibitory concentration for vancomycin greater than 8 µg/mL, and it was identified as a vancomycin-resistant Staphylococcus. This result was consistent with Pournajaf et al.’s [32] finding that vancomycin resistance was 2.5%, and this figure was lower than that from a review from Ethiopia, where the pooled prevalence of VRSA was 5.3% [17], as well as the findings from Debre Markos (14.1%) [33] and elsewhere (29.4%) [34].

From all methicillin-resistant Staphylococci, the *mecA* gene was carried by 27.5% of the isolates. This finding was comparable with a study from Nigeria, where 30.5% of the isolates carried the *mecA* gene [29]. In the present study, 12.9% of *S. aureus* carried the *mecA* gene, which is lower than studies reported in Ethiopia (20%) [35], Bangladesh (25%) [36], Nigeria (38%) [29], and Iran (45.1%) [32]. It should be noted that the majority of isolates exhibiting the *mecA* gene were discovered in Debre Tabor. Eight (72.7%) of the ten *mecA*-positive isolates were detected at Debre Tabor Hospital. The reason might be poor socioeconomic status, personal demographics, antibiotic prescription practice, and infection control practices, which are associated with increased MRSA infection rates [30,31].

In our present study, the *femA* gene was detected in all *S. aureus* isolates, except two cefoxitin-resistant strains (6.7%). This finding was comparable with a study from China [37]. Additionally, in the present study, the *femA* gene was found in *S. haemolyticus*, *S. warneri*, and 83.3% of *M. sciuri* cefoxitin-resistant strains. On the other hand, neither *mecA* nor *femA* were detected in *S. epidermidis* [37]. The primers used should be specific to *S. aureus*; therefore, it is somewhat surprising that the two *S. aureus* lack the gene and that several other non-aureus isolates carry the gene. The explanation could be mutational changes in *S. aureus* and gene transfer to other species. All of these isolates have been sent for whole genome sequencing, and this matter will be analyzed further when the results are ready. 

It is interesting that a significantly higher proportion of CoNS isolates harbour methicillin resistance genes, where 83.3% of *M. sciuri* and 50% of *S. haemolyticus* carried the *mecA* gene. This is in agreement with early reports that CoNS were the most common species in nosocomial infections and exhibit higher antibiotic resistance rates than *S. aureus.* This may be explained by the high prevalence of methicillin resistance linked with staphylococcal cassette chromosome (SCCmec) elements in CoNS [38], and they are considered a major reservoir of SCCmec [39]. For instance, Berglund et al. described the likely transfer of a type V SCCmec from methicillin-resistant *S. haemolyticus* to MSSA, thus transforming into MRSA [9,40]. Another study revealed that the *mecA* homologue in *M. sciuri* may be an evolutionary precursor to MRSA pathogenic strains, highlighting the main routes of antibiotic resistance gene transfer [41]. Furthermore, the report demonstrated that MSSA become MRSA by acquiring SCCmec from *S. epidermidis* through horizontal transfer [42]. These accounts suggest that horizontal interspecies transfer of mobile genetic elements could be a crucial element for MRSA global dissemination [40,41,43].

The absence of the *mecA* and *vanA* genes in the MRSA and VRSA samples does not imply the absence of resistance, as resistance may be due to other mutations or cassette-containing resistance genes [44]. Globally, resistant staphylococcal isolates lacking the *mecA* gene show the possibility for additional mechanisms to compete with *mecA* in the establishment of MRSA [45,46]. MRSA’s resistance against beta-lactams and methicillin is further complicated by its ability to develop resistance to vancomycin through accidental transmission of the *vanA* gene from Enterococcal strains [47]. Vancomycin is a glycopeptide antibiotic that prevents the formation of the peptidoglycan layer by binding to the peptide precursor. Antibiotic overuse leads to bacterial resistance, thus prompting the search for new antimicrobial strategies [48]. Genomics can identify antibiotic targets, and live non-multiplying bacteria can be targeted for new antibacterials, potentially resulting in new antibacterials that shorten therapy microorganisms, reduce adverse effects, and potentially reduce antibacterial resistance [49]. Preclinical research explores metal uptake via bacterial metallophores [48]. Bacteriophages have been demonstrated to be antibacterial in animals that are susceptible to certain infectious diseases [49].

In the present study, the likelihood of methicillin-resistant staphylococcus SSI increased among patients aged ≥ 61 years (*p* = 0.025, AOR = 3.729 (1.179–11.791)). Similar findings have been reported in Brazil [22]. Previous studies have suggested that patients on antibiotics (*p* = 0. 017), who had a previous wound infection (*p* = 0.006), and with a hospital stay > 72 h showed an association with MRSA infection [33]. Similarly to our finding, previous use of antibiotics (*p* = 0.025, AOR = 3.066 (1.101–9.392)) and preoperative hospital stays > 7 days (*p* = 0.000, AOR = 1.856 (0.688–5.311)) demonstrated an association with methicillin-resistant Staphylococci for SSI. Unlike our study, a report by X. Yang et al. [50] showed that long, invasive procedures used in the ICU, such as tracheal intubation and ventilator usage, along with patients with cerebral infarction and other embolisms increase the likelihood of developing MRSA colonization and further infections.

## 4. Materials and Methods

### 4.1. Study Area and Design

A cross-sectional study was conducted in four purposively selected University Teaching Hospitals, including Debre Tabor Comprehensive Specialized Hospital (DTCSH), Tikur Anbessa Specialized Hospital (TASH), Hawassa University Teaching Hospital (HUTH), and Jimma University Teaching Hospital (JUTH) in Ethiopia. These hospitals provide a range of services in both outpatient and inpatient units under different wards, such as general surgical, gynecology, obstetric/maternity, and orthopedics, and they all have microbiology laboratories for culture and antimicrobial sensitivity testing. This study was conducted between July 2020 and August 2021.

### 4.2. Variables

The variables in this study were MRSA and VRSA infections, socio-demographic characteristics, clinical data, and risk factor variables, such as age, sex, surgical site, length of hospital stay, history of hospital admission, previous use of antibiotics, smoking history, alcohol consumption, type and nature of the surgery, type of antimicrobial prophylaxis, history of previous antibiotic use, surgical procedure performed, and duration of the operation.

### 4.3. Study Population and Sampling

The study population consisted of patients admitted for elective and emergency surgery in general surgery, gynecology/obstetric, and orthopedics wards. All surgical patients, regardless of their age, who underwent surgery during the study period and developed signs and symptoms of surgical site infection (SSI) within 30 days were included in this study. Consent and/or assent was secured from each participant before the commencement of data collection. Patients who developed SSIs after 30 days following the operation, those who refused to participate, patients with infected burn wounds, and those on treatment were excluded from this study. 

### 4.4. Sample Size and Sampling Technique

A total of 752 clinically diagnosed cases of SSI from different wards were enrolled in this study. The sample size was calculated based on a single population proportion sample size estimation formula (n = Z^2^ P (1 − P)/d^2^) using a proportion (P) of 20% [51]. As this was a multicenter study, to increase the sample size, a precision (d) of 0.03 was used, where Z stands for Z statistic with a confidence level of 95% and a Z value of 1.96. Considering a 10% non-response rate, the final total sample size was estimated at 752. Enrollments continued until the necessary sample size was achieved, with proportional allocation among the different hospitals based on patient flow.

### 4.5. Specimen Collection, Isolation, and Identification of S. aureus

Wound swabs or aspirates were collected based on standard operation procedure (SOP). Conventional bacteriological techniques, such as morphological, cultural, and biochemical characterization, were used to identify strains of *S. aureus* [52]. The specimens were inoculated on blood agar plates (BAP) (Oxoid, UK), and mannitol salt agar (MSA) (Oxoid) and then incubated at 35 °C for 24 h. The *S. aureus* isolates were identified through Gram staining, catalase and coagulase tests, including golden yellow colonies on MSA, which were considered phenotypic identification tests.

### 4.6. Identification Confirmation of the Species of Bacteria Strain Using Matrix-Assisted Laser Desorption/Ionization Time-of-Flight Mass Spectrometry (MALDI-TOF MS)

All phenotypically cefoxitin-resistant *S. aureus* isolates were re-identified using MALDI-TOF MS [53] at the Clinical Microbiology Department, Uppsala University Hospital in Sweden. A single colony of bacteria from fresh cultures was smeared onto a MALDI-TOF plate, air-dried, treated with formic acid and MALDI matrix solution, and again air-dried before reading. MALDI-TOF identification scores were automatically generated by the system software [54], and isolates with scores of two and above were accepted, while those with scores below 1.7 and flagged red were rejected. Samples with scores between 1.7 and 2 and flagged yellow were re-analyzed.

### 4.7. Antimicrobial Susceptibility Testing

Antimicrobial susceptibility testing (AST) was carried out using the cefoxitin disc diffusion test, which is a surrogate marker test for oxacillin resistance following the clinical and laboratory standard institute (CLSI) protocol [55], and the minimum inhibitory concentration (MIC) of the vancomycin strip was determined using the E-test method on MHA. The reference strain *S. aureus* (ATCC^®^ 25923, Seattle, DC, USA) was used as a quality control. Evidence showed that MRSA is a requisite for VISA [56]. Hence, we screened VRSA/VISA from MRSA isolates.

### 4.8. Identification of Methicillin-Resistant S. aureus Strains

#### 4.8.1. DNA Extraction

DNA was extracted from all cefoxitin-resistant *S. aureus* isolates through the boiling method, as described previously [57]. Briefly, each isolate was grown overnight on nutrient agar (Oxoid, UK), and 3 to 5 colonies of that culture were suspended in 300 µL of 1× Tris EDTA buffer. The suspension was subjected to 10 min of boiling at 94 °C in a water bath (Thermo Fisher Scientific, CA, USA), followed by 10 minutes of freezing at −20 °C, 1 min at room temperature, and 5 min of centrifugation at 14,000× *g*. Finally, 150 µL of the supernatant was transferred into a nuclease-free Eppendorf tube and measured using Nanodrop (Thermo Scientific) for the quality and quantity of DNA prior to storage at −20 °C until analysis.

#### 4.8.2. Standardization of Multiplex PCR for the Detection of *Staphylococci mecA*, *femA*, *vanA*, and *vanB*

Multiplex PCR was used to amplify different genes that are associated with methicillin resistance from Staphylococci. The primers and annealing temperatures were standardized for the detection of *S. aureus mecA*, *vanA*, *vanB*, and *femA*. The PCR products were analyzed through gel electrophoresis. Positive and negative control strains were included in all amplification reactions to ensure accuracy of the test results.

First, PCR was standardized using a range of annealing temperatures to establish the optimum annealing reaction condition for all of the primers. All PCR primers are described in Table 5. The reaction mixture contained 12.5 μL of hot star master mix (Qiagen, Hilden, Germany), 0.5 μL each of the forward and reverse primers, 9 μL of molecular-grade water, and 2.5 μL of the template, with a final volume of 25 μL. Amplification for *mecA* and *femA* was carried out over 40 cycles of initial heat activation at 95 °C for 15 min, denaturation at 94 °C for 30 s, followed by annealing at 52 °C for 45 s, extension at 72 °C for 1 min, and final extension at 72 °C for 10 min. Amplification for *vanA* and *vanB* was carried out over 40 cycles of initial heat activation at 95 °C for 15 min, denaturation at 94 °C for 30 s, followed by annealing at 56 °C for 90 s, extension at 72 °C for 90 min, and final extension at 72 °C for 10 min. The PCR products were analyzed through electrophoresis on a 2% agarose gel and detected through staining in ethidium bromide with the aid of a gel imaging system, GelDoc (Bio-Rad). The following controls were included in all amplification reactions: ATCC 33591 (*mecA*-positive *S. aureus*) and ATCC 25923 (*mecA*-negative *S. aureus*).

**Table 5 antibiotics-12-01681-t005:** Primers used in multiplex PCR for the detection of the *mecA*, *vanA*, *vanB*, and *femA* genes.

Target Gene	Primer Name	Primer Sequence (5′-3′)	Size bp	References
*mecA*	MF	GTAGAAATGACTGAACGTCCGATAA	310	[58]
	MR	CCAATTCCACATTGTTTCGGTCTAA		
*vanA*	VF	GGGAAAACGACAATTGC	732	[59]
	VR	GTACAATGCGGCCGTTA		
*vanB*	VF	ACCTACCCTGTCTTTGTGAA	300	
	VR	AATGTCTGCTGGAACGATA		
*femA*	FF	AAAAAAGCACATAACAAGCG	132	[60]
	FR	GATAAAGAAGAAACCAGCAG		

### 4.9. Quality Assurance 

Specimens were collected according to the recommended standard operating procedures (SOPs). The performance of all prepared culture media (BAP and MSA) was also checked by inoculating control strains, *S. aureus* (ATCC^®^ 25923), for each new batch of agar plates prepared. In addition, the sterility of culture media was checked by incubating 5% of the prepared media at 37 °C for 24–48 h. In addition, reagents for Gram stain and biochemical tests were checked against control strains of *S. aureus.* The 0.5 McFarland standard was used to standardize the bacterial suspension inoculum density for the susceptibility test. Each MALDI-TOF run also included *S. aureus* (ATCC^®^ 25923) as a quality control strain. Furthermore, the performance of the antibiotic disks was evaluated using American-type cell culture (ATCC) controls. As such, *S. aureus* ATCC^®^ 25923 (cefoxitin zone 21–29 mm) and *S. aureus* ATCC^®^ 43300 (zone ≤ 21 mm) were used as control strains to determine the performance of the cefoxitin disc diffusion test. *S. aureus* ATCC^®^ 29213 MIC of vancomycin broth value 0.5–2.0 µg/mL was used as a control strain to measure the performance of vancomycin [55].

### 4.10. Data Entry and Analysis

The data were checked for completeness, missing values, and coding of questionnaires entered into the Research Electronic Data Capture (RED-Cap). A double data entry method was used to ensure the accuracy of the data, and data were analyzed using STATA version 25. Descriptive statistics were used to present antimicrobial susceptibility patterns. Frequencies and cross-tabulations were used to summarize descriptive statistics. Logistic regression was used to study the effect of independent variables on the dependent variables. *p*-values less than 0.05 were considered statistically significant. 

### 4.11. Ethical Considerations 

The Department of Medical Microbiology, Immunology, and Parasitology (DMIP) and the AHRI/ALERT Research Ethics Committee (AAREC) reviewed and approved this study. Institutional review board (IRB) approval was also obtained from Addis Ababa University’s College of Health Sciences, AAUMF03-008/2020. Selected hospitals received a formal letter from the AHRI and DMIP, and each hospital’s medical directors gave their consent. Written consent/assent was taken from each study participant before initiation of the actual data collection.

Patient information was kept confidential by sharing the laboratory results of research participants only with the designated accountable clinicians. Patients who experienced SSIs were managed according to hospital policy. In general, this study was conducted in accordance with the Declaration of Helsinki.

## 5. Conclusions

A multicenter study identified 11 mecA-positive Staphylococci species, with 36.4% being MRSA, but no VRSA was found among these MRSA. What is more captivating in this study is a significantly high prevalence of *mecA* carriage among CoNS, suggesting difficulties in the treatment of patients with CoNS infections. Furthermore, this signifies a huge potential of MSSA conversion to MRSA through horizontal gene transfer, which would make things more complicated. In terms of geographic distribution, out of 11 *mecA*-gene-positive Staphylococci, 8 (72.7%) were detected in DTCSH, with significant variations between hospitals, suggesting that strategies to control methicillin-resistant Staphylococci should be tailored to specific hospitals. The presence of staphylococcal isolates was linked to factors like older age, hospital stay, antibiotic history, and prophylaxis. Prompt prevention and control measures for MRSA-high-risk populations, including strict adherence to infection prevention methods, periodic surveillance, and antibiotic stewardship programs, are crucial for effective treatment and prevention strategies. 

## Figures and Tables

**Figure 1 antibiotics-12-01681-f001:**
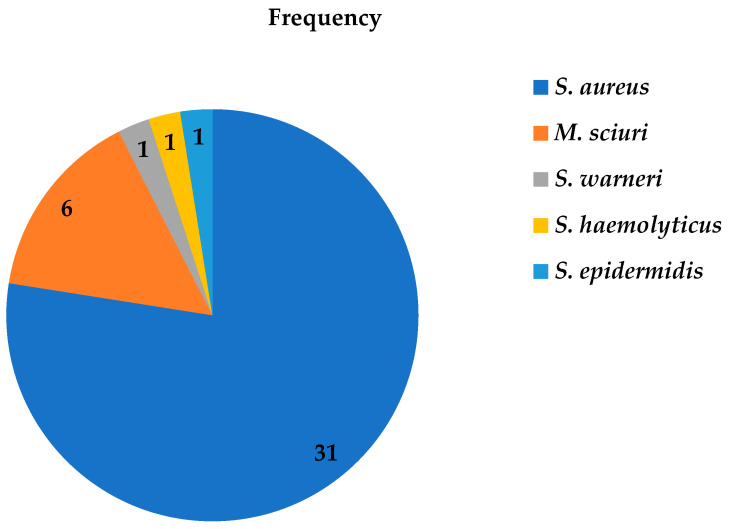
Frequency of methicillin-resistant Staphylococci isolates from patients diagnosed with surgical site infection at four different hospitals in Ethiopia using MALDI-TOF MS between July 2020 and August 2021.

**Figure 2 antibiotics-12-01681-f002:**
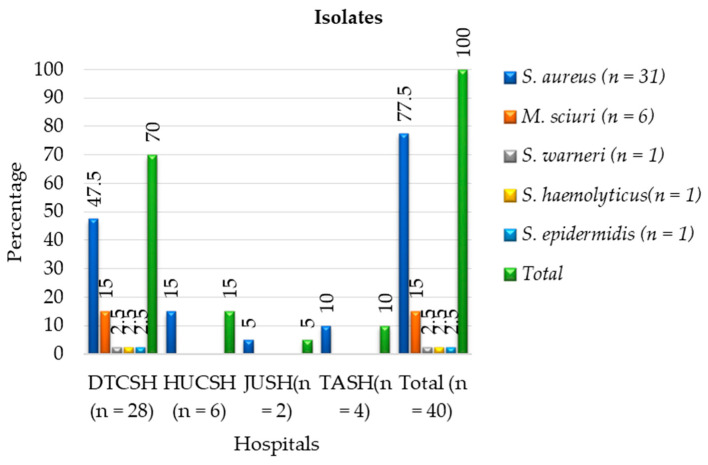
Frequencies of MALDI-TOF MS identification and distribution of phenotypic methicillin-resistant Staphylococci and *M. sciuri* isolates at each hospital between July 2020 and August 2021. DTCSH: Debre Tabor Comprehensive Specialized Hospital; HUCSH: Hawassa University Comprehensive Specialized Hospital; JUTSH: Jimma University Teaching Specialized Hospital; TASH: Tikur Anbessa Specialized Hospital.

**Figure 3 antibiotics-12-01681-f003:**
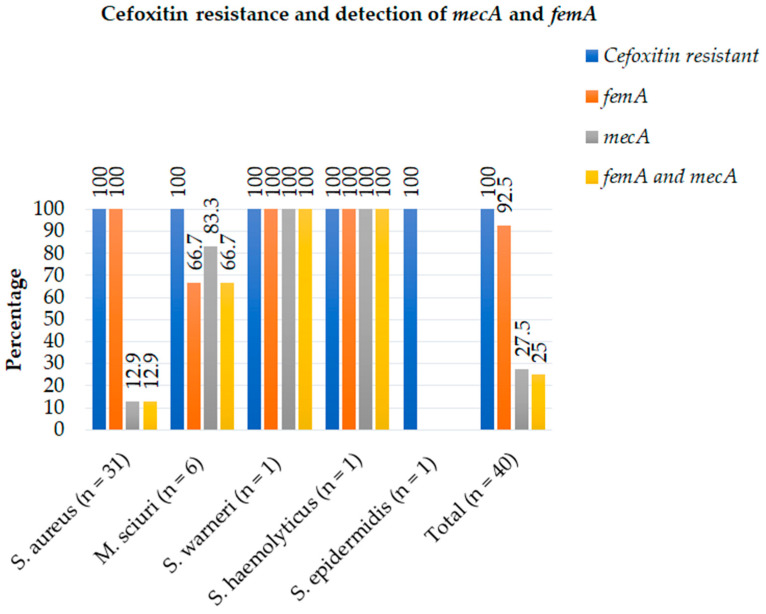
Frequency and distribution of cefoxitin resistance and PCR-confirmed gene among each Staphylococci isolate in patient diagnosed with surgical site infection between July 2020 and August 2021.

**Figure 4 antibiotics-12-01681-f004:**
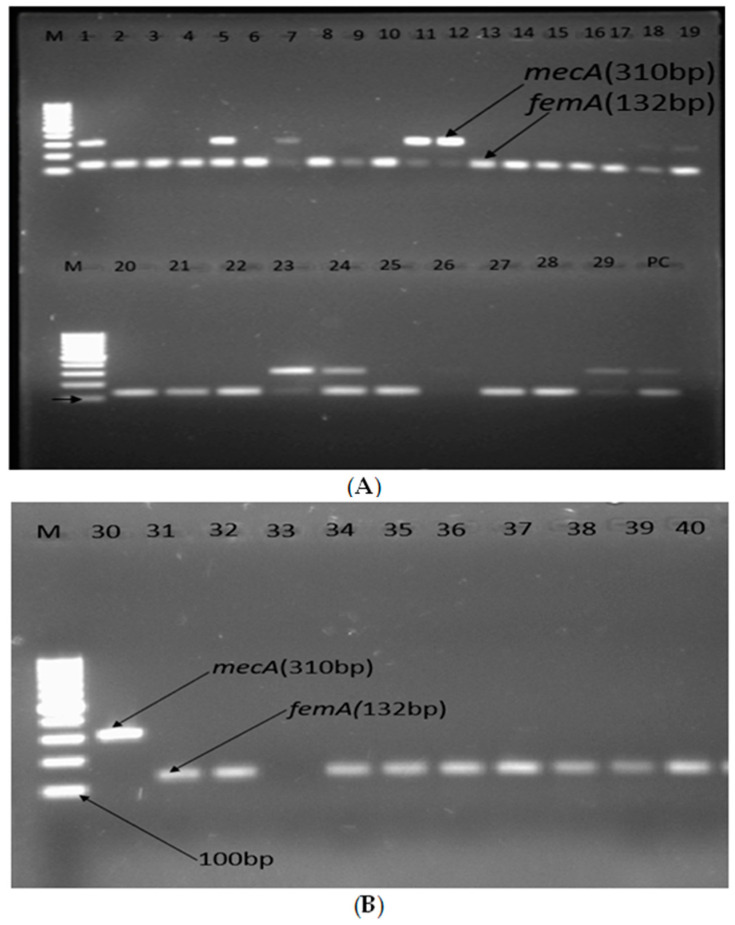
(**A**,**B**) Agarose gel electrophoresis showing bands of *femA* and *mecA* genes of methicillin-resistant Staphylococcic strains from patients diagnosed with surgical site infection at four different hospitals in Ethiopia; lane M1: 100 bp molecular weight ladder; lane PC: positive control; lanes 1–40 are tested isolates, and positive amplification of *femA* and *mecA* is indicated by 132 bp and 310 bp PCR amplicons, respectively.

**Figure 5 antibiotics-12-01681-f005:**
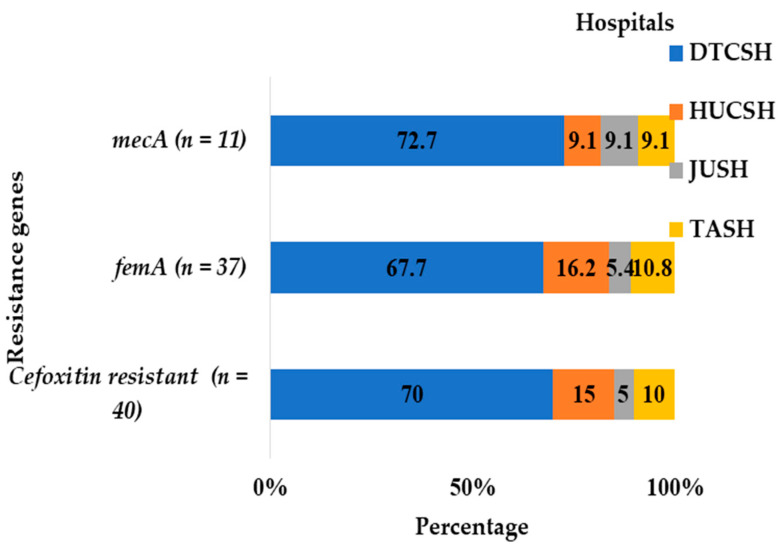
Frequency and distribution of cefoxitin-resistant isolates and *mecA* and *femA* genes from the total number of Staphylococci and *M. sciuri* isolates at each hospital between July 2020 and August 2021. DTCSH: Debre Tabor Comprehensive Specialized Hospital; HUCSH: Hawassa University Comprehensive Specialized Hospital; JUTSH: Jimma University Teaching Specialized Hospital; TASH: Tikur Anbessa Specialized Hospital.

**Table 1 antibiotics-12-01681-t001:** Antibiotic resistance pattern of *S. aureus* isolates from patients diagnosed with surgical site infection at four different hospitals in Ethiopia between July 2020 and August 2021.

Antibiotics	Methods	N (%), % = N/752	N (%), % = N/163	AST Results	Strain
**Cefoxitin**	I-Z ≥ 22 mm	123 (16.4)	123 (75.5)	S	MSSA
	≤21 mm	40 (5.3)	40 (24.5)	R	MRSA

Abbreviations: S, susceptible; R, resistant; I, intermediate; I-Z, inhibition zone; MSSA, methicillin-sensitive *S. aureus*; MRSA, methicillin-resistant *S. aureus*; AST, antimicrobial susceptibility test.

**Table 2 antibiotics-12-01681-t002:** Socio-demographic characteristics and clinical data of *S. aureus* among patients diagnosed with SSI at four different hospitals in Ethiopia from July 2020 to August 2021.

Variables/Characteristics		Frequency of *S. aureus* (%)
**Gender**	Male	95 (58.3)
	Female	68 (41.7)
**Age in (years)**	≤18	25 (20.9)
	19–40	77 (54)
	41–60	19 (13.5)
	≥61	42 (11.7)
**Surgical site infection**	Superficial	79 (48.5)
	Deep	84 (51.5)
**Preoperative hospital stays**	<7	77 (47.2)
	≥7	86 (52.8)
**Previous use of antibiotics**	Yes	79 (48.5)
	No	84 (51.5)
**Smoking**	Yes	16 (9.8)
	No	147 (90.1)
**Alcoholic**	Yes	48 (29.4)
	No	115 (70.6)
**Nature of surgery**	Emergency Elective	55 (68.1) 108 (31.9)
**Type of surgery**	Clean/Clean contaminated surgery	148 (90.8)
	Contaminated surgery	15 (9.2)
**Timing of surgical antimicrobial prophylaxis**	Before the operation	59 (36.2)
	During the operation	104 (63.8)
**Duration of operation**	<1 h	100 (52.8)
	≥1 h	63 (47.2)

**Table 3 antibiotics-12-01681-t003:** Bivariate and multivariate analysis for identification of methicillin resistance Staphylococci predictors among patients diagnosed with surgical site infection at four different hospitals in Ethiopia from July 2020 to August 2021.

Characteristics		Bacterial Growth		*p*-Value	Crude-OR (95%CI)	Adjusted-OR (95%CI)	*p*-Value
		MRSA	MSSA				
**Gender**	Male	29 (17.8)	66 (40.5)	0.039	2.276 (1.0444–4.9633)	1.638 (0.597–4.489)	0.337
	Female	11 (6.7)	57 (35)			1	
**Age in (years)**	≤18	2 (1.2)	23 (14.1)	0.000	2.788 (1.8716–4.154)	0.556 (0.1014–3.046)	0.499
	19–40	11 (6.7)	66 (40.5)			1	
	41–60	2 (1.2)	17 (10.4)			1.556 (0.259–9.328)	0.628
	≥61	25 (15.3)	17 (10.4)			3.729 (1.179–11.791)	**0.025**
**Preoperative hospital stays**	≤7	13 (8)	64 (39.3)	0.034	2.253 (1.064–4.771)	1	
	>7	27 (16.7)	59 (36.2)			1.856 (0.688–5.311)	**0.000**
**Previous use of antibiotics**	Yes	26 (16)	53 (32.5)			3.692 (1.059–2.800)	**0.025**
	No	14 (8.9)	70 (42.9)	0.001	3.256 (1.724–7.634)	1	
**History of alcohol intake**	Yes	18 (11)	30 (18.4)			1.075 (0.1331–8.6925)	0.945
	No	22 (13.5)	93 (57.1)	0.015	2.536 (1.202–5.351)	1	
**Nature of surgery**	Elective	16 (9.8)	92 (56.4)			1	
	Emergency	24 (14.7)	31 (19)	0.000	4.452 (2.098–9.445)	1.962 (0.0619–6.224)	0.000
**Timing of surgical antimicrobial prophylaxis**	Before the operation	7 (4.3)	57 (35)			1	
	After the operation	33 (20.2)	71(43.6)	0.006	3.453 (2.098–9.445)	3.066 (1.001–9.392)	0.05
**Duration of operation**	≤1 h	19 (11.7)	81 (49.7)			1	
	>1 h	21 (12.9)	42 (25.8)	0.004	2.132 (1.034–4.396)	1.890 (0.6321–5.652)	0.235

Bold: *p*-value statistically significant association.

**Table 4 antibiotics-12-01681-t004:** Presentation of the cefoxitin and vancomycin resistance patterns of *mecA* carrying Staphylococci, and the distribution of *femA* and *van* genes among patients diagnosed with SSI at four different hospitals in Ethiopia from July 2020 to August 2021.

Lane (*mecA*pos)	MALDI-TOF MS	Study Site	Cefoxitin	Vancomycin	*femA*	*mecA*	*vanA* and *vanB*
1	*S. aureus*	DTCSH	R	S	Pos	Pos	Neg
5	*S. aureus*	JUTSH	R	S	Pos	Pos	Neg
7	*M. sciuri*	DTCSH	R	S	Pos	Pos	Neg
11	*S. haemolyticus*	DTCSH	R	R	Pos	Pos	Neg
12	*S. warneri*	DTCSH	R	S	Pos	Pos	Neg
18	*S. aureus*	TASH	R	S	Pos	Pos	Neg
19	*S. aureus*	HUCSH	R	S	Pos	Pos	Neg
23	*M. sciuri*	DTCSH	R	S	Pos	Pos	Neg
24	*M. sciuri*	DTCSH	R	S	Pos	Pos	Neg
29	*M. sciuri*	DTCSH	R	S	Pos	Pos	Neg
31	*M. sciuri*	DTCSH	R	S	Neg	Pos	Neg

## Data Availability

The data sets generated during and/or analyzed during the current study are available from the corresponding authors upon reasonable request. The data are not publicly available due to privacy restrictions.

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
