# Peer review of "Molecular Epidemiology of Methicillin-Resistant Staphylococcus aureus among Patients Diagnosed with Surgical Site Infection at Four Hospitals in Ethiopia"

_antibiotics, 2023, doi:10.3390/antibiotics12121681_

Round 1
Reviewer 1 Report
Comments and Suggestions for Authors
The manuscript “Phenotypic and Molecular Epidemiology of Methicillin-Resistant Staphylococcus aureus among patients diagnosed with 3 Surgical Site Infection in Ethiopia” is a very well-designed study. However, there are some of the suggested improvements for enhanced interests and to further improve the quality of the manuscript.
Title: The title could be modified as “Molecular Epidemiology of Methicillin-Resistant Staphylococcus aureus among patients diagnosed with 3 Surgical Site Infection in Ethiopia”.
1. Line 123: it should be “standard operating procedure”
2. Line 236-237: The proportion of MRSA became 40/752 (5.3%) ------ this line needs revision
3. Line 249 and Line 259: Table 3 and Table 4, there should be similar variables among both tables. Further, is smoking or alcohol could be considered as a risk factor for SSI? The word “SEX” as risk factor could be modified as “Gender”
4. Line 261-266: The paragraph is not clear regarding non- S. aureus. Are the isolates M. sciuri or other species of staphylococci shown methicillin resistant? However, the data in Figure-2 suggests to the readers that all 40 are methicillin resistant.
5. CoNS: CoNS were only discussed in the “Discussion”, there number could also be mentioned in the results somewhere
6. Lines 288-292. The data showed that Hospital DTCSH have maximum numbers of MRSA. If there were some specific reasons which could be described in the discussion or in the risk factors section, should be added or considered.
Minor mistakes of English Grammar need attention in the manuscript.
Finally, the manuscript could be considered after these suggested modifications/ suggestions.
Thank and Regards
Comments on the Quality of English Language
The quality of English is up to mark for understanding the manuscript.
Author Response
Response to reviewer
- Summary
Thank you very much for taking the time to review this manuscript. Please find the detailed responses below and the corresponding revisions/corrections highlighted/in track changes in the re-submitted files.
This is a well-written, interesting paper. Minor revisions could be considered:
Thank you very much,
The manuscript “Phenotypic and Molecular Epidemiology of Methicillin-Resistant Staphylococcus aureus among patients diagnosed with 3 Surgical Site Infection in Ethiopia” is a very well-designed study. However, there are some of the suggested improvements for enhanced interests and to further improve the quality of the manuscript.
Title: The title could be modified as “Molecular Epidemiology of Methicillin-Resistant Staphylococcus aureus among patients diagnosed with 3 Surgical Site Infection in Ethiopia”.
- Comment accepted and modified as “Molecular Epidemiology of Methicillin-Resistant Staphylococcus aureus among Patients Diagnosed with Surgical Site Infection at Four Hospitals in Ethiopia”(line no 1)
- Line 123: it should be “standard operating procedure”
- Comment accepted and corrected (line no 127)
- Line 236-237: The proportion of MRSA became 40/752 (5.3%)------ this line needs revision
- Comment accepted and changed to: Of all participants, 5.3% (40/752) were MRSA, while among isolates of aureus, it was 24.5%. (Line no 227-229).
- Line 249 and Line 259: Table 3 and Table 4, there should be similar variables among both tables. Further, is smoking or alcohol could be considered as a risk factor for SSI?
- No, because in this instance, a p-value of less than 0.02 in bivariate analysis and less than 0.05 in multivariable analysis is required for independent variables to be regarded as risk factors.
- The word “SEX” as risk factor could be modified as “Gender”
- Accepted and modified as Gender (line no 241 and 252)
- Line 261-266: The paragraph is not clear regarding non- aureus. Are the isolates M. sciurior other species of staphylococci shown methicillin resistant? However, the data in Figure-2 suggests to the readers that all 40 are methicillin resistant.
- Yes, phenotypically all 40 were methicillin-resistant including those CoNS ( sciuri, S. haemoliticus, and S. warneri) and modified as Figure 2. MALDI- TOF MS frequency identification and distribution of phenotypic methicillin resistance (line no 264)…Staphylococci and M. sciuri isolates
- Additionally to show mecA Fig3
Fig 3. Frequency and distribution of cefoxitin resistance and PCR confirmed gene among each Staphylococci isolate in patient diagnosed with surgical site infection between July 2020 and August 2021.(line no 281-283)
- CoNS: CoNS were only discussed in the “Discussion”, there number could also be
- Sorry there is in the abstract, introduction and result part (as Staphylococci isolate ) (Line 26, 41-46,264)
- Lines 288-292. The data showed that Hospital DTCSH have maximum numbers of MRSA. If there were some specific reasons which could be described in the discussion or in the risk factors section, should be added or considered.
- Accepted and described in the discussion part, please
On the other hand, the majority of isolates exhibiting the mecA gene were discovered in Debre Tabor. Eight (72.7%) of the ten mecA gene-positive isolates were detected in Debre Tabor Hospital. The Variations might be poor socioeconomic status, personal demographics, antibiotic prescription practice, and infection control practices are associated with increased MRSA infection rates (1). (line no 350-354)
Please minor mistakes of English Grammar need attention in the manuscript.
- Accepted and revised, please
With regards,
Response to reviewer
- Summary
Thank you very much for taking the time to review this manuscript. Please find the detailed responses below and the corresponding revisions/corrections highlighted/in track changes in the re-submitted files.
This is a well-written, interesting paper. Minor revisions could be considered:
The manuscript “Phenotypic and Molecular Epidemiology of Methicillin-Resistant Staphylococcus aureus among patients diagnosed with 3 Surgical Site Infection in Ethiopia” is a very well-designed study. However, there are some of the suggested improvements for enhanced interests and to further improve the quality of the manuscript.
Title: The title could be modified as “Molecular Epidemiology of Methicillin-Resistant Staphylococcus aureus among patients diagnosed with 3 Surgical Site Infection in Ethiopia”.
- Comment accepted and modified as “Molecular Epidemiology of Methicillin-Resistant Staphylococcus aureus among Patients Diagnosed with Surgical Site Infection at Four Hospitals in Ethiopia”(line no 1)
- Line 123: it should be “standard operating procedure”
- Comment accepted and corrected (line no 127)
- Line 236-237: The proportion of MRSA became 40/752 (5.3%)------ this line needs revision
- Comment accepted and changed to: Of all participants, 5.3% (40/752) were MRSA, while among isolates of aureus, it was 24.5%. (Line no 227-229).
- Line 249 and Line 259: Table 3 and Table 4, there should be similar variables among both tables. Further, is smoking or alcohol could be considered as a risk factor for SSI?
- No, because in this instance, a p-value of less than 0.02 in bivariate analysis and less than 0.05 in multivariable analysis is required for independent variables to be regarded as risk factors.
- The word “SEX” as risk factor could be modified as “Gender”
- Accepted and modified as Gender (line no 241 and 252)
- Line 261-266: The paragraph is not clear regarding non- aureus. Are the isolates M. sciurior other species of staphylococci shown methicillin resistant? However, the data in Figure-2 suggests to the readers that all 40 are methicillin resistant.
- Yes, phenotypically all 40 were methicillin-resistant including those CoNS ( sciuri, S. haemoliticus, and S. warneri) and modified as Figure 2. MALDI- TOF MS frequency identification and distribution of phenotypic methicillin resistance (line no 264)…Staphylococci and M. sciuri isolates
- Additionally to show mecA Fig3
Fig 3. Frequency and distribution of cefoxitin resistance and PCR confirmed gene among each Staphylococci isolate in patient diagnosed with surgical site infection between July 2020 and August 2021.(line no 281-283)
- CoNS: CoNS were only discussed in the “Discussion”, there number could also be
- Sorry there is in the abstract, introduction and result part (as Staphylococci isolate ) (Line 26, 41-46,264)
- Lines 288-292. The data showed that Hospital DTCSH have maximum numbers of MRSA. If there were some specific reasons which could be described in the discussion or in the risk factors section, should be added or considered.
- Accepted and described in the discussion part, please
On the other hand, the majority of isolates exhibiting the mecA gene were discovered in Debre Tabor. Eight (72.7%) of the ten mecA gene-positive isolates were detected in Debre Tabor Hospital. The Variations might be poor socioeconomic status, personal demographics, antibiotic prescription practice, and infection control practices are associated with increased MRSA infection rates (1). (line no 350-354)
Please minor mistakes of English Grammar need attention in the manuscript.
- Accepted and revised, please
With regards,
Reviewer 2 Report
Comments and Suggestions for Authors
This is a well-written, interesting paper. Minor revisions could be considered:
1. Mamallicoccus sciuri: please check the spelling which should be Mammaliicoccus sciuri
2. A few more information about this uncommon bacterium would be appreciated like … it is part of the normal flora of goats and camels while its role in human diseases is still uncertain…
3. At line 321 it could be better to specify “… in an Indian hospital (39).”
4. The prevalence of MRSA in Ethiopia in this study is 24.5% while all other previous papers found a much higher prevalence between 32.5, 68.4 and 82.3%. This is a striking difference showing that MRSA is not yet dominant in all Ethiopian hospitals. Comments from the authors who know well their health system would highly valued.
5. Malditof spectrometry was the only test done at Uppsala University Hospital but then the Swedish author G Swedberg should be listed among the laboratory investigators (Author contribution).
As a clinician, I would have appreciated if MRSA in Ethiopia were sensitive to cotrimoxazole and doxycycline which are widely used in their countryside; however, this information could be better part of another study.
Author Response
Response to reviewer
Summary
Thank you very much for taking the time to review this manuscript. Please find the detailed responses below and the corresponding revisions/corrections highlighted/in track changes in the re-submitted files.
This is a well-written, interesting paper. Minor revisions could be considered:
- Mamallicoccus sciuri: please check the spelling which should be Mammaliicoccus sciuri
- Accepted and corrected as Mammaliicoccus sciuri (line no 26 and 44).
- A few more information about this uncommon bacterium would be appreciated like … it is part of the normal flora of goats and camels while its role in human diseases is still uncertain…
- Accepted and incorporated (line no 45)
- At line 321 it could be better to specify “… in an Indian hospital (39).
- Accepted and modified (line no 321)
- The prevalence of MRSA in Ethiopia in this study is 24.5% while all other previous papers found a much higher prevalence between 32.5, 68.4, and 82.3%. This is a striking difference showing that MRSA is not yet dominant in all Ethiopian hospitals. Comments from the authors who know well their health system would highly valued.
- Variations in MRSA prevalence across countries are influenced by demographics, antibiotic prescription policies, infection prevention and control programs, staff and elderly hygiene education, healthcare system structure, and MRSA diagnostic facilities (line no 336)
- Malditof spectrometry was the only test done at Uppsala University Hospital but then the Swedish author G Swedberg should be listed among the laboratory investigators (Author contribution).
- I am sorry YS, MY, and AAl are only listed under laboratory investigation even though he participated in subculturing and DNA extraction for MALDI TOF MS analysis and WGS and analysis, interpretation of the findings but now I included his contributions (line no 429)
As a clinician, I would have appreciated if MRSA in Ethiopia were sensitive to cotrimoxazole and doxycycline which are widely used in their countryside; however, this information could be better part of another study.
- I appreciate your feedback, and we will investigate this more. I asked information internists directly about how they treat MRSA, and they informed me that doxycycline and cotrimoxazole are prescribed, though not frequently.
With regards,
Reviewer 3 Report
Comments and Suggestions for Authors
2.2 Variables
The division of factors into dependent and independent variables seems unnecessary.
2.5 Specimen Collection, Isolation, and Identification of S. aureus
Gram-positive, catalase-positive, slide coagulase-positive cocci that form golden yellow colonies on MSA do not confirm the presence of S. aureus.
2.7 Antimicrobial Susceptibility Testing
There is no need to describe the details of the methodology for performing a drug susceptibility test, which exists in the CLSI recommendations. Such as, for example, applying Etest strips to the medium.
There was an error in the title of Table 1. Instead of "Primers used in multiplex PCR for the detection of the mecA, vanA, vanB and famA genes" there is "Primers used in multiplex PCR for the detection of the mecA, vanA, vanB and famA genes July 2020 to August 2021".
2.9 Quality Assurance
Reference strain numbers should be standardized, e.g. ATCC® 25923 instead of ATCC 25923 or ATCC25923.
3. Results
In Table 3, sometimes there is a space before the bracket and other times it is not.
Line 265: Instead of "with (47.5%) as S. aureus" it should be "with 47.5% as S. aureus".
Line 271: There is an error in the title of Figure 2 (instead of MALDI TOF-MS there is MLDI TOF-MS).
Line 306: In the title of Table 5, cefoxitin and vancomycin are incorrectly capitalized.
Additionally, there are errors in Table 5, e.g. column Lane (what does it refer to?), MALDTOF-MS, femA, mecA gene (should be without gene), vanA&Bgenes (incorrect nomenclature).
References
Line 472: error in cited references - CaLSI?
Furthermore, many times the names of bacteria are not written in italics.
Author Response
Summary
Thank you very much for taking the time to review this manuscript. Please find the detailed responses below and the corresponding revisions/corrections highlighted/in track changes in the re-submitted files.
2.2 Variables
The division of factors into dependent and independent variables seems unnecessary.
- Comments accepted and modified by removing dependent and independent, only listing variables (line no.102)
2.5 Specimen Collection, Isolation, and Identification of S. aureus
Gram-positive, catalase-positive, slide coagulase-positive cocci that form golden yellow colonies on MSA do not confirm the presence of S. aureus
- Yes, it is not confirmatory test comment accepted and corrected as a kind of phenotypic identification method of S. aureus. (line no 133)
2.7 Antimicrobial Susceptibility Testing
There is no need to describe the details of the methodology for performing a drug susceptibility test, which exists in the CLSI recommendations. Such as, for example, applying Etest strips to the medium
- Comment accepted and modified as follows (line no 146-149)
The antimicrobial susceptibility testing (AST) was carried out by using cefoxitin disc diffusion test, which is a surrogate marker test for oxacillin resistance following the clinical and laboratory standard institute (CLSI) protocol (1) and the minimum inhibitory concentration (MIC) of vancomycin strip was determined using the Etest method on MHA.
There was an error in the title of Table 1. Instead of "Primers used in multiplex PCR for the detection of the mecA, vanA, vanB and famA genes" there is "Primers used in multiplex PCR for the detection of the mecA, vanA, vanB and famA genes July 2020 to August 2021".
- Comments accepted and corrected (line no 186).
- Table 1. Primers used in multiplex PCR for the detection of the mecA, vanA, vanB and femAgenes (line no 186).
2.9 Quality Assurance
Reference strain numbers should be standardized, e.g. ATCC® 25923 instead of ATCC 25923 or ATCC25923.
- Comment accepted and corrected as ATCC® 25923 (line no 189-199)
- Results
In Table 3, sometimes there is a space before the bracket and other times it is not.
- Comment accepted and corrected a space before the bracket
Line 265: Instead of "with (47.5%) as S. aureus" it should be "with 47.5% as S. aureus".
- Comment accepted corrected as 47.5% (line no 258)
Line 271: There is an error in the title of Figure 2 (instead of MALDI TOF-MS there is MLDI TOF-MS).
- Comment accepted and corrected as MALDI-TOF MS (line no 264).
Line 306: In the title of Table 5, cefoxitin and vancomycin are incorrectly capitalized.
- Comment accepted replaced with small letters (cefoxitin and vancomycin) (line no 316).
Additionally, there are errors in Table 5, e.g. column Lane (what does it refer to?), MALDTOF-MS, femA, mecA gene (should be without gene), vanA&Bgenes (incorrect nomenclature).
- Comment accepted and corrected as below
Column Lane refers to mecA positive and MALDI-TOF MS, femA, mecA, vanA and vanB (line no 318)
References
Line 472: error in cited references - CaLSI?
- Comment accepted and corrected as Clinical Laboratory Standard Institute (line no 500)
Furthermore, many times the names of bacteria are not written in italics.
- Comment accepted and italics
With regards
Reviewer 4 Report
Comments and Suggestions for Authors
Dear Authors,
Your Article entitled "Phenotypic and Molecular Epidemiology of Methicillin-Resistant Staphylococcus aureus among patients diagnosed with Surgical Site Infection in Ethiopia" has been reviewed,
This paper deserves attention since it highlights on a very important topic related to the Epidemiology of MRSA among patients suffering from SSIs in Ethiopia, in fact this study was done using molecular and medical characterization of the bacterial isolates from infected sites. Such kind of studies has a great potential effect on Public Health of the population living in Ethiopia, on both Community and Hospital levels.
The Article is well presented and written, some English and Design Modifications are required.
Kindly find below the list of my comments (Minor and Major Ones):
01- In the Whole manuscript, you are kindly asked to use the same font and the same font size. Example, in the Abstract section, between lines 25 and 27, different fonts and font sizes were used, this must be homogenous.
02- In the Keywords section, Authors are invited to remove duplicates, example (Methicillin-resistant Staphylococci and MRSA) are the same. So you are invited to put the following keywords:
Surgical Site Infection, Methicillin-resistant Staphylococci, Molecular Epidemiology, Ethiopia, and Antimicrobial Resistance.
03- In the Introduction section, Line 38, you are invited to remove the term "superbug" and replace it by another scientific term.
04- In the Whole manuscript, the first time you use the name of a bacterium, you are kindly asked to put its abbreviation between parenthesis, then you can use just the abbreviation in the text. Example: Line 38, Staphylococcus aureus (S. aureus).
05- In the Introduction section, When you talked about the virulence of S. aureus, you are invited to list some of its virulence factors including Toxins, Enzymes, Metallophores (siderophores), and Resistance to antibiotics:
Kindly add the following articles as references for this point:
Reference 01: Pathogenicity and virulence of Staphylococcus aureus.
Reference 02: The Key Element Role of Metallophores in the Pathogenicity and Virulence of Staphylococcus aureus: A Review
06- Concerning the paragraph entitled "Sample Size and Sampling Technique", regarding the calculation of the Sample Size of the study, you are kindly requested to put an article containing the prevalence of SSI in Ethiopia or a country near by Ethiopia as a reference for the calculation of this number.
You can see if you can use this article as reference for this point:
Reference 03: Surgical site infection and its associated factors in Ethiopia: a systematic review and meta-analysis
07- In the Whole manuscript when you write the name of a bacterial gene, you are invited to put it in small letter and in italic, Example: Lines 172-173, you are kindly asked to replace "famA, vanA and vanB" by "femA (not famA), vanA and vanB".
08- In the Table 1, when authors put the primers used for the PCR, why they are using two reference for the gene femA? I suggest to remove one of these two references.
09- In the Figure 3, you are kindly asked to put the name of genes in small letter and in italic.
10- Regarding the Figure 4, you put two figures under the name (A.), you are kindly asked to put them as A. and B.
11- In the Discussion section, you are kindly invited to talk about the importance of the discovery or the creation of new families of antibiotics.
You can use the following articles as references for this idea:
Reference 04: Novel approaches to developing new antibiotics for bacterial infections
Reference 05: Towards new antibiotics classes targeting bacterial metallophores
12- In the References section, Line 545, Reference 53., you are kindly asked to put Author's name in English.
Best Regards,
Comments on the Quality of English LanguageDear Authors,
The Article is well written in English, but some sentences need paraphrazing.
Best Regards,
Author Response
Summary
Thank you very much for taking the time to review this manuscript. Please find the detailed responses below and the corresponding revisions/corrections highlighted/in track changes in the re-submitted files.
Your Article entitled "Phenotypic and Molecular Epidemiology of Methicillin-Resistant Staphylococcus aureus among patients diagnosed with Surgical Site Infection in Ethiopia" has been reviewed,
This paper deserves attention since it highlights on a very important topic related to the Epidemiology of MRSA among patients suffering from SSIs in Ethiopia, in fact, this study was done using molecular and medical characterization of the bacterial isolates from infected sites. Such kind of studies has a great potential effect on the Public Health of the population living in Ethiopia, on both Community and Hospital levels.
The Article is well presented and written, some English and Design Modifications are required.
Kindly find below the list of my comments (Minor and Major Ones):
01- In the Whole manuscript, you are kindly asked to use the same font and the same font size. Example, in the Abstract section, between lines 25 and 27, different fonts and font sizes were used, this must be homogenous.
- Accepted and corrected
02- In the Keywords section, Authors are invited to remove duplicates, example (Methicillin-resistant Staphylococci and MRSA) are the same. So you are invited to put the following keywords:
Surgical Site Infection, Methicillin-resistant Staphylococci, Molecular Epidemiology, Ethiopia, and Antimicrobial Resistance.
- Accepted and corrected by additionally, Molecular Epidemiology, and Antimicrobial Resistance. (line no 36).
03- In the Introduction section, Line 38, you are invited to remove the term "superbug" and replace it by another scientific term
- Accepted and replaced with cocci.(line no 39).
04- In the Whole manuscript, the first time you use the name of a bacterium, you are kindly asked to put its abbreviation between parenthesis, then you can use just the abbreviation in the text. Example: Line 38, Staphylococcus aureus (S. aureus).
- Accepted and included in bracket Staphylococcus aureus ( aureus).(line no 38)
05- In the Introduction section, When you talked about the virulence of S. aureus, you are invited to list some of its virulence factors including Toxins, Enzymes, Metallophores (siderophores), and Resistance to antibiotics:
Kindly add the following articles as references for this point:
Reference 01: Pathogenicity and virulence of Staphylococcus aureus.
Reference 02: The Key Element Role of Metallophores in the Pathogenicity and Virulence of Staphylococcus aureus: A Review
- Comments accepted and incorporate in the introduction part used both references (line no 45-50)
- aureus possesses a unique set of virulence factors, including toxins, enzymes, and metallophores, which enable it to survive extreme conditions, promote tissue colonization, and cause systemic infection (1). By utilizing metallophores, this bacterium can sequester metal ions from its environment (2).
06- Concerning the paragraph entitled "Sample Size and Sampling Technique", regarding the calculation of the Sample Size of the study, you are kindly requested to put an article containing the prevalence of SSI in Ethiopia or a country near by Ethiopia as a reference for the calculation of this number.
- Dear reviewer I already put an article containing the 20% prevalence of SSI in Ethiopia (Ref no.17) as follows Sample size estimation formula (n= Z2 P (1 - P) /d2) using a proportion (P) of 20% (3).(line no 121)
You can see if you can use this article as reference for this point:
Reference 03: Surgical site infection and its associated factors in Ethiopia: a systematic review and meta-analysis
07- In the Whole manuscript when you write the name of a bacterial gene, you are invited to put it in small letter and in italic, Example: Lines 172-173, you are kindly asked to replace "famA, vanA and vanB" by "femA (not famA), vanA and vanB".
- Comment accepted and corrected with small letters and in italic femA, vanA, and vanB (line no 187).
08- In the Table 1, when authors put the primers used for the PCR, why they are using two references for the gene femA? I suggest to remove one of these two references.
- Comment accepted and removed one (line no 187).
09- In the Figure 3, you are kindly asked to put the name of genes in small letter and italics.
- Comment accepted and corrected with small letters and in italic mecA and femA and modified as Figure3 in to 4 (line no 293)
10- Regarding the Figure 4, you put two figures under the name (A.), you are kindly asked to put them as A. and B.
- Comment accepted and corrected levelled with A and B (line no 304-305)
11- In the Discussion section, you are kindly invited to talk about the importance of the discovery or the creation of new families of antibiotics.
- Comment accepted and incorporated (line no 388-394)
- Antibiotic overuse leads to bacterial resistance, prompting the search for new antimicrobial strategies (5). Genomics can identify antibiotic targets, and live non-multiplying bacteria can be targeted for new antibacterials, potentially resulting in new antibacterials that shorten therapy microorganisms, reduce adverse effects, and potentially reduce antibacterial resistance (4). Preclinical research explores metal uptake via bacterial metallophores (5). Bacteriophages have been demonstrated to be antibacterial in animals that are susceptible to certain infectious diseases (4).
You can use the following articles as references for this idea:
Reference 04: Novel approaches to developing new antibiotics for bacterial infections
Reference 05: Towards new antibiotics classes targeting bacterial metallophores
12- In the References section, Line 545, Reference 53., you are kindly asked to put Author's name in English.
- Comment accepted corrected Author's name in English as Hawra Wahab Aziz A (line no 569)
With Regards,
- Cheung GYC, Bae JS, Otto M. Pathogenicity and virulence of Staphylococcus aureus. Virulence. 2021;12(1):547-69.
- Ghssein G, Ezzeddine Z. The Key Element Role of Metallophores in the Pathogenicity and Virulence of Staphylococcus aureus: A Review. Biology (Basel). 2022;11(10).
- Mengesha RE, Kasa BG, Saravanan M, Berhe DF, Wasihun AG. Aerobic bacteria in post surgical wound infections and pattern of their antimicrobial susceptibility in Ayder Teaching and Referral Hospital, Mekelle, Ethiopia. BMC Res Notes. 2014;7:575.
- Ezzeddine Z, Ghssein G. Towards new antibiotics classes targeting bacterial metallophores. Microbial Pathogenesis. 2023;182:106221.
(4)
Round 2
Reviewer 4 Report
Comments and Suggestions for Authors
Dear Authors,
Your revised version of this work was reviewed,
I would like to thank you for all the modifications you made,
The article is better for publication in its present form,
I have just one remark concerning the corrections you made in my comment number 11, you did not add this article to your references:
Reference 05: Towards new antibiotics classes targeting bacterial metallophores.
You are kindly invited to add this reference.
Best Regards,
Author Response
Dear reviewer,
Thank you very much for taking the time to review this manuscript. Please find the detailed responses below and the corresponding revisions/corrections highlighted/in track changes in the re-submitted files.
Comments and Suggestions for Authors
Your revised version of this work was reviewed,
I would like to thank you for all the modifications you made,
Thank you,
The article is better for publication in its present form,
I have just one remark concerning the corrections you made in my comment number 11, you did not add this article to your references:
Reference 05: Towards new antibiotics classes targeting bacterial metallophores.
- Dear reviewer I agree with your important comment, please
- Comment accepted and reference incorporated (line no 388-394) from the document reference no 56.
- Antibiotic overuse leads to bacterial resistance, prompting the search for new antimicrobial strategies (1). Genomics can identify antibiotic targets, and live non-multiplying bacteria can be targeted for new antibacterials, potentially resulting in new antibacterials that shorten therapy microorganisms, reduce adverse effects, and potentially reduce antibacterial resistance (2). Preclinical research explores metal uptake via bacterial metallophores (1). Bacteriophages have been demonstrated to be antibacterial in animals that are susceptible to certain infectious diseases (2).
With regards,
References
- Ezzeddine Z, Ghssein G. Towards new antibiotics classes targeting bacterial metallophores. Microbial Pathogenesis. 2023;182:106221.
- Coates AR, Hu Y. Novel approaches to developing new antibiotics for bacterial infections. Br J Pharmacol. 2007;152(8):1147-54.
